# Beyond the Apnea–Hypopnea Index: Exploring Time-Dependent Hazard Ratios of Respiratory Events in Obstructive Sleep Apnea

**DOI:** 10.3390/arm93050046

**Published:** 2025-10-16

**Authors:** Wojciech Kuczyński, Aleksandra Kudrycka, Karol Pierzchała, Izabela Grabska-Kobyłecka, Michael Pencina, Sebastian Sakowski, Piotr Białasiewicz

**Affiliations:** 1Department of Sleep Medicine and Metabolic Disorders, Medical University of Lodz, Mazowiecka 6/8, 92-215 Lodz, Poland; aleksandra.kudrycka@umed.lodz.pl (A.K.); karol.pierzchala@stud.umed.lodz.pl (K.P.); piotr.bialasiewicz@umed.lodz.pl (P.B.); 2Department of Clinical Physiology, Medical University of Lodz, Mazowiecka 6/8, 92-215 Lodz, Poland; izabela.grabska-kobylecka@umed.lodz.pl; 3Department of Biostatistics and Bioinformatics, Duke Clinical Research Institute, Duke University, 300 West Morgan St., Durham, NC 27701, USA; michal.pencina@duke.edu; 4Faculty of Mathematics and Computer Science, University of Lodz, Stefana Banacha 22, 90-238 Lodz, Poland; sebastian.sakowski@wmii.uni.lodz.pl; 5Centre for Data Analysis, Modelling and Computational Sciences, University of Lodz, Narutowicza 68, 90-136 Lodz, Poland

**Keywords:** obstructive sleep apnea (OSA), apnea–hypopnea index (AHI), mortality, polysomnography (PSG)

## Abstract

**Highlights:**

**What are the main findings?**
The highest long-term mortality risk was associated with severe OSA, particularly events occurring during REM sleep and significant oxygen desaturation. Dynamic modeling revealed temporal patterns of risk progression, providing insights beyond those captured by static, point-in-time analyses.

**What is the implication of the main finding?**
REM-related apneas and oxygen desaturation measures emerge as stronger predictors of long-term outcomes than conventional AHI-based metrics.

**Abstract:**

Obstructive sleep apnea (OSA) is associated with increased risks of systemic comorbidities, leading to significant morbidity and mortality. This study investigates predictors of all-cause mortality, emphasizing the interplay of clinical symptoms, polysomnographic findings, and comorbidities. The aim of this study was to identify and compare respiratory predictors of all-cause mortality over 5, 10, and 15 years. A single-center study was conducted at a Sleep Medicine Department between 2005 and 2019, 4025 patients with suspected OSA who underwent polysomnography were admitted, 853 died during the study. We performed Cox regression analyses with dynamic hazard ratios to evaluated predictors of mortality. Prevalence of OSA was high—75.6% in the cohort: 929 patients with mild OSA (23.1%), 770 with moderate OSA (19.1%), and 1343 with severe OSA (33.4%). Survival rates were 89.7%, 81.9%, and 78.8% at 5, 10, and 15 years, respectively. Cardiovascular causes dominated mortality (33.3%), followed by cancer (26.5%). AHI_REM_ was associated with higher mortality risk in 0–5, 0–10, 0–15 years of observation in contrast to AHIN_REM_ and AHI_TST_. The hazard ratio analysis showed that mortality risk changed over time depending on sleep stage and event type: risk increased for AHI_REM_ and AHI_TST_, while it stayed the same or decreased for AHI_NREM_ and most central apneas.

## 1. Introduction

Obstructive sleep apnea (OSA) is characterized by the recurrent cessation of breathing (apneas) or partial upper airway obstruction (hypopneas) during sleep and affects from 10% to 30% of the population [1,2]. OSA increases the risk of morbidity and mortality and has been recognized as a risk factor that leads to premature death among patients with arterial hypertension, obesity, diabetes, and cardiovascular diseases [3,4,5] Polysomnography (PSG) is considered the gold standard in OSA diagnostics, allowing for the identification and classification of apneas and hypopneas. The apnea–hypopnea index (AHI) is a well-known, widely used metric for indicating the severity of OSA [6]. There is growing evidence that the AHI does not accurately estimate the risk of death, even though the AHI is used to diagnose and evaluate the severity of OSA [7,8]. Depending on the numbers of apneas and hypopneas per hour, OSA can be classified as mild (AHI ≥ 5, to <15), moderate (AHI ≥ 15 to <30), or severe (AHI ≥ 30) [6]. The AHI typically refers to the total sleep time (TST). However, it is important to remember that AHI can be based on the time spent sleeping in REM and NREM sleep, referred to as AHI_REM_ and AHI_NREM_, respectively. Additionally, the AHI can be calculated for the back or side positions, referred to as AHI_back_ and AHI_side_, respectively. Therefore, categorization relying solely on the AHI related to TST proves inadequate. Consequently, extensive analyses have been conducted, leading to the identification of distinct phenotypic manifestations of OSA. Most of analyses consider the AHI as the sole predictor of disease severity, often neglecting critical data such as the predominant type of respiratory events or OSA subgroups related to sleep positions or sleep stages. We hypothesize that individuals with OSA have diverse predictors of all-cause mortality beyond the AHI. Although the impact on morbidity and mortality and the precise relationship between OSA and mortality remains complex and multifaceted, the mortality among OSA patients can be attributed to different respiratory events.

## 2. Materials and Methods

### 2.1. Study Design

This was a retrospective, observational, single-center cohort study conducted at the Department of Sleep Medicine and Metabolic Disorders, the Medical University of Lodz. Patients were admitted between 2005–2019; all of them underwent overnight PSG, which was one of inclusion criteria. Patients that were referred with a suspected diagnosis of OSA were admitted to the Sleep Clinic at 20:00 and underwent physical examination (with measurement of body mass, height, calculated body mass index (BMI), heart rate, and blood pressure). All patients completed a questionnaire regarding comorbidities with the attending physician: arterial hypertension, diabetes, atrial fibrillation, dyslipidemia, depression, stroke, myocardial infraction, and drugs (statins, oral hypoglycemic drugs, insulin, acetylsalicylic acid (ASA), anticoagulants). Signs and symptoms of OSA were self-reported by the patients during the admission with special interest in sleepiness self-reported by the patients and measured using the Epworth Sleepiness Scale (ESS). In the case of any doubts, the attending physician clarified the questions and verified the list of medications taken. Other symptoms, including morning fatigue, snoring, and morning headaches, were self-reported by the participants. Patients were classified as having arterial hypertension if: (1) self-reported a history of arterial hypertension, or (2) self-reported medications used in the treatment of arterial hypertension, or (3) the mean blood pressure exceeded 140/90 mm Hg during one of three measurements (first consultation prior to the polysomnographic examination, blood pressure measurement before or immediately after the polysomnographic examination). Dyslipidemia was defined as self-reported dyslipidemia by the patient or the presence of statins in the list of medications taken regularly. Diabetes was defined as self-reported diabetes or the presence of hypoglycemic agents, including insulin. All study variables are presented in Table 1.

### 2.2. Outcome Variables

Patients were followed for up to 15 years for all-cause mortality and plausibly OSA-related mortality (Figure 1). The cause of death was verified by the Ministry of Digital Affairs and the Central Statistical Office in accordance with International Classification of Diseases (ICD-10). Mortality data were recorded up to 24 February 2024. Loss to follow-up was negligible, as mortality status was obtained for all patients through comprehensive national registries, see Table 2.

### 2.3. Polysomnography

PSG started at 22:00 till 7:00 next day; somnographic findings were collected from the inpatients reports. A standard nocturnal, one-night polysomnography was performed by recording the following channels: electroencephalography (C4\A1, C3\A2, F3\O1 F4\O2), chin muscles and anterior tibialis electromyography, electrooculography, measurements of oronasal air flow (a nasal canula), snoring, body position (a gravitational gauge placed on the sternum), respiratory movements of chest and abdomen (piezoelectric gauges), unipolar electrocardiogram, and average saturation; time with saturation < 90% (SpO_2_ < 90) denotes oxygen saturation level as measured by pulse oximetry (Sleep Lab, Jaeger–Viasys, Hoechberg, Germany). Sleep stages were scored manually by the three somnologists with the European Sleep Research Society certification according to the criteria based on a 30-s epoch standard. Two primary types of respiratory events are distinguished in PSG: apneas and hypopneas. An apnea is defined as a complete cessation of airflow or a reduction in its amplitude by more than 90% from baseline, lasting at least 10 s. Based on the underlying mechanism, apneas are classified as obstructive, central, or mixed. Obstructive apneas are characterized by continued respiratory effort, as indicated by thoracic and abdominal movements, whereas central apneas are marked by the absence of respiratory muscle activity. Mixed apneas begin with a central component, followed by obstructive features. Hypopneas are defined as a reduction in airflow amplitude by more than 30%, accompanied by either a ≥ 3% drop in oxygen saturation (SpO_2_) or an arousal recorded on electroencephalography (EEG) [6,7]. According to recent guidelines, PSG assessment also includes the identification of central mediated hypopneas, allowing for a more comprehensive understanding of the pathophysiology of sleep-disordered breathing [9]. T90 (time with desaturation below 90%) was defined as the cumulative percentage of total sleep time spent with oxygen saturation below 90%, as measured by pulse oximetry. Total sleep time (TST) was defined as the sum of all sleep stages (NREM and REM) expressed in hours, and AHI_TST_ was calculated as the total number of apneas and hypopneas divided by TST. Electroencephalography arousals were scored according to the American Academy of Sleep Medicine Guidelines. The diagnosis of OSA was according to guidance of the International Classification of Sleep Disorders 2 [6]. All patients who received the results of the polysomnographic examination were given detailed recommendations regarding the need to initiate treatment in accordance with current guidelines: weight reduction, mandibular advancement devices (MAD), positional therapy, and CPAP therapy.

### 2.4. Statistics

The R Studio software (version 2023.12.1) was used for data analysis. Analyses used the following R packages (versions): survival 3.8.3, finalfit 1.1.0, ranger 0.17.0, ggplot2 4.0.0, dplyr 1.1.4, ggfortify 0.4.19, tidyverse 2.0.0, openxlsx 4.2.8, forcats 1.0.0, survminer 0.5.0, readxl 1.4.5, and dynamichazard 1.0.2. We fitted multivariable Cox proportional hazards models to identify prognostic factors for mortality, reporting hazard ratios (HRs) with 95% confidence intervals (CI). Continuous variables were tested for normality with Shapiro–Wilk and summarized as mean ± SD when normal and min–max when not normal. Categorical data are *n* (%) and compared using chi-square or Fisher’s exact tests, as appropriate. Kaplan–Meier curves display survival over 0–15 years, with groups compared by the log-rank test. OSA severity (mild, moderate, severe) was modeled with no OSA as the reference. Variables were grouped a priori into four sets: (1) age, (2) clinical signs/symptoms and anthropometrics (weight, height, BMI, blood pressure); (3) comorbidities and medications; and (4) polysomnographic indices (see Table 3). We also performed univariable Cox analyses; detailed results are provided in Appendix A Table A1. The proportional hazards (PHs) assumption was evaluated for each covariate using Schoenfeld residuals with the cox.zph function from the survival package. The global and variable-specific tests did not indicate significant violations (all *p* > 0.05). In addition to the standard Cox proportional hazards model, we applied a dynamic hazard model implemented in the R package dynamichazard (function ddhazard) to explore and visualize potential time-dependent effects. This approach treats the regression coefficients as a time-evolving state vector following a random-walk process, while the baseline hazard is approximated by a piecewise constant function over a predefined time grid.

This study was conducted in compliance with the amended Declaration of Helsinki, and the Ethics Committee of the Medical University of Lodz approved the study protocol (RNN/23/15/KE; RNN/393/19/KE). Patient consent was waived due to the retrospective type of study. The study was neither funded by an institutional grant nor pharmaceutical industry or medical company. This manuscript was developed with the support of the Medical Research Agency as part of an educational program: Polish Clinical Scholars Research Training realized by Harvard Medical School Postgraduate Medical Education, Boston, Massachusetts. The manuscript is the authors’ work and is not affiliated with Harvard Medical School or the Medical Research Agency.

## 3. Results

A total of 4644 patients met the inclusion criteria (3157 men, 72.1%). A total of 474 patients were excluded from the analysis: 220 patients had a total sleep time below 150 min, 292 patients had crucial variables of interest missing, 74 patients were excluded due to time of observation over 15 years, and 35 patients were excluded due to BMI and age below 18 (see Figure 1). The study group of 4023 patients was included in the statistical analysis. Among them, 982 patients (24.4%) were referred with a presumptive diagnosis of OSA based on typical symptoms but did not meet the diagnostic criteria for OSA (AHI < 5) and served as the control group. During 15 years of observation 3170 patients (78.80%) remained alive and 853 (21.20%) became deceased; clinical characteristics of the group are listed in Table 1. The predominant causes of death over the 15 years of follow-up were cardiovascular (284, 33.3%), cancer (226, 26.5%), and pulmonary (102, 12%). Based on the International Classification of Diseases (ICD-10), the main causes included: chronic heart failure I.50 (97, 21.1%), myocardial infraction I.21–I.25 (72, 15.6%), stroke, intracerebral hemorrhage I.60–I.64 (62, 13.5%), and chronic obstructive pulmonary disease J.44 (52, 11.3%). The detailed ICD-10 diagnosis codes were not included in this study; however, they may be provided upon reasonable request to the corresponding author. Five-year survival rate was 89.7%, 10-year survival rate was 81.9%, and 15-year survival was 78.8%.

Counts of participants by follow-up interval (based on OS_time_years) were: 0–5 years: 1041 (25.9%), >5–10 years: 1911 (47.5%), and >10–15 years: 1071 (26.6%). The Kaplan-Meier survival curves mortality over a 15-year follow-up period illustrate significant differences (*p* < 0.001) in survival probabilities based on the severity of obstructive sleep apnea (OSA), Figure 2. The prevalence of OSA, defined as AHI ≥ 5 and accompanied by typical symptoms, was high—75.6% in the cohort. This included 929 patients with mild OSA (23.1%), 770 with moderate OSA (19.1%), and 1343 with severe OSA (33.4%).

The results of the Cox regression analyses revealed factors associated with both all-cause and OSA-related mortality over three time periods (up to 5, 10 and 15 years). Univariate analyses showed significant associations with mortality across all timeframes among study variables. Some of the variables had no impact on mortality risk among study groups, i.e., clinical symptoms (snoring, morning headaches) and polysomnography findings (AHI < 5) (see Appendix A Table A1).

In the multivariate analyses among clinical signs and symptoms, sleepiness assessed by the ESS was associated significantly with mortality (0–5 years HR 1.28, 95% CI: 1.16–1.42; 0–10 years HR 1.20, 95% CI: 1.11–1.30; 0–15 years HR 1.17, 95% CI:1.09–1.26), see Table 3. In the group of comorbidities, episodes of stroke were associated with mortality (0–5 years HR 2.41, 95% CI: 1.66–3.49; 0–10 years HR 1.53, 95% CI: 1.17–1.99; 0–15 years HR 1.77, 95% CI: 1.38–2.28). In the group of medications, use of new oral anticoagulants was associated with mortality (0–5 years HR 2.56, 95% CI: 1.67–3.94; 0–10 years HR 2.07, 95% CI: 1.44–2.98; 0–15 years HR 1.71, 95% CI: 1.21–2.43). OSA severity was associated with mortality in the third analyzed group of polysomnographic findings. AHI_REM_ was associated with significantly higher mortality (0–5 years HR 1.38, 95% CI: 1.18–1.62; 0–10 years HR 1.26, 95% CI: 1.12–1.41; 0–15 years HR 1.23, 95% CI: 1.11–1.37). There was no significant association between AHI_NREM_ and mortality between study groups, see Figure 3.

The dynamic analysis of hazard ratio (HR) trajectories over a 15-year period revealed distinct temporal patterns; the most pronounced and persistent increases in HR over time were observed for the following. AHI_REM_: consistently elevated HR across the 15-year timeframe, in line with point estimates (HR 1.38; 95% CI: 1.18–1.62; *p* < 0.001). AHI_TST_: showed a progressive increase in HR, reaching statistical significance in later periods (HR 1.27; 95% CI: 1.17–1.37; *p* < 0.001), despite non-significance in early years. Obstructive apneas during REM: demonstrated a steady upward trend in HR, supported by significant point estimates (HR 1.38; 95% CI: 1.26–1.51; *p* < 0.001). Severe OSA: exhibited the strongest and most time-dependent effect, with HR increasing markedly over time (up to HR 4.72; 95% CI: 3.40–6.56; *p* < 0.001), underscoring a robust dose–response relationship. In contrast, AHI_NREM_ and central apneas during REM showed minimal or non-significant changes in HR over time, which aligns with their non-significant point estimates (*p* > 0.1), see Figure 4.

Additionally, we performed dynamic hazard ratio analyses for our variables to gain a deeper understanding of their relationships.

## 4. Discussion

Our findings highlight the importance of incorporating a broader spectrum of clinical and polysomnographic data to improve patient outcomes and reduce mortality risk. This study offers detailed insights into comorbidities and chronic medication use in patients referred for sleep disorder evaluation, enhancing the understanding of OSA complexity. The prevalence of depression and suicides (even higher than car accidents) among patients referred to sleep disorder clinics reflects the multifaceted nature of their health issues. Notably, depression was associated with a reduced risk of all-cause mortality (HR 0.66, 95% CI: 0.45–0.96, *p* = 0.031), underscoring the necessity of diagnostics and treatment within interdisciplinary clinical teams. Patients with depressive disorders may demonstrate heightened concern for their overall health and increased sensitivity to alarming symptoms. This phenomenon is not restricted to the domain of mental health but may also extend to somatic conditions. Such individuals are often more vigilant toward bodily sensations, which may lead to earlier recognition of symptoms and proactive health-seeking behaviors. Several mechanisms may underlie this tendency, including somatization, health-related anxiety, and heightened interoceptive awareness commonly observed in depression. While this vigilance may facilitate earlier detection and intervention in the course of various diseases, it may also contribute to increased healthcare utilization and a potential overestimation of health risks. Future studies should further investigate the extent to which depressive symptomatology influences health perception and the timing of medical intervention across different patient populations. While AHI is a well-established metric for diagnosing and evaluating the severity of OSA, it fails to capture the full spectrum of risk factors associated with mortality [5]. Our findings align with the existing literature suggesting that the complexity of OSA and its impact on health cannot be adequately summarized by the AHI alone. Factors such as sleepiness according to the ESS, REM and non-REM sleep disturbances, and a range of comorbid conditions significantly influence mortality risk [5]. The increased mortality associated with severe OSA and duration time of desaturation < 90% highlights the importance of nocturnal hypoxemia as a critical mediator of adverse outcomes [10]. Interestingly, our findings demonstrate a complex interaction between respiratory events during REM and NREM sleep, AHI_REM_, AHI_NREM_, AHI_TST_, and saturation.

### Polysomnographic Findings

The association of AHI_REM_ with mortality in early follow-up (0–5 years) and AHI_NREM_ in longer-term follow-up suggests that apneas occurring in different sleep stages may have distinct physiological impacts. This aligns with evidence that REM apneas are associated with greater hypoxemia and sympathetic activation, whereas NREM apneas contribute to chronic volume overload and cardiovascular strain [11,12]. The persistence of AHI_TST_ as a predictor of mortality at 0–15 years underscores the cumulative impact of overall apnea burden on long-term outcomes. These findings highlight the utility of polysomnography not only for diagnosis but also for prognostic stratification in OSA patients.

The survival curves demonstrate a clear dose–response relationship between OSA severity and mortality. Severe OSA patients consistently exhibited the worst outcomes, which aligns with prior research linking untreated or poorly managed OSA to cardiovascular disease, systemic inflammation, and metabolic dysfunction [10,13]. Patients with mild OSA had survival probabilities like those without OSA, suggesting that mild OSA may have a relatively limited impact on long-term mortality. Moderate OSA, however, marked the transition to more significant mortality risks, emphasizing the importance of early intervention to prevent progression to severe OSA [14,15]. The patterns for OSA-related mortality were like those for all-cause mortality, indicating that OSA’s systemic effects contribute broadly to both direct (e.g., respiratory failure) and indirect (e.g., cardiovascular disease) mortality risks. This underscores the need to address comorbid conditions, such as hypertension, diabetes, and dyslipidemia, in OSA management. Our analysis, in contrast to the work of Azarian et al., focused on the impact of comorbidities, medications, and medical history rather than age as an independent prognostic factor. The results of our study provide a broader perspective on the complexity of OSA [16].

To explore the topic of dynamic hazard ratios (HRs) in obstructive sleep apnea (OSA), particularly in relation to the variations of risk factors over time and across different clinical contexts, several pertinent studies have been identified. These studies illustrate the significance of utilizing dynamic modeling approaches to better understand the impact of OSA on mortality and other health-related outcomes. For instance, Kim et al. (2020) [17] examined various polysomnographic characteristics associated with OSA and their implications on mortality. They noted that metrics such as the ratio of REM to NREM apneas could indicate physiological differences that might influence hazard ratios over time, emphasizing the importance of understanding how different patterns of sleep disturbances correlate with long-term outcomes [17,18]. Collapsing follow-up into three arbitrary time bins forces a step-function for the effect of each exposure and discards within-bin variation. This reduces power, induces residual confounding within bins, and can even reverse or attenuate effects depending on where cut-points fall. Dynamic models treat time continuously (typically with splines), using all events and preserving dose–time information [19]. If an exposure’s effect changes as follow-up accrues, a proportional-hazards model summarized by three time frames mis-specifies the effect and spreads it unevenly across bins [20]. Pathophysiologic burdens in OSA—especially REM-dominant obstructive events and hypoxic burden—are episodic and cumulative, not static. Dynamic HRs therefore track the evolving risk signal more faithfully than bin-averaged HRs. Evidence shows REM-related OSA is strongly linked to adverse cardiometabolic outcomes, and that hypoxic burden (and related desaturation metrics) is a robust predictor of cardiovascular events and mortality, signaling your dynamic HRs are captured more cleanly [12,21].

Stronger, persistent signals for REM-linked metrics (obstructive apnea in REM) in the dynamic curves mirror known biology (greater sympathetic surges and hypoxemia clustering in REM) and previously reported associations with cardiometabolic risk. Interval HRs dilute this pattern by averaging across long windows, especially if events concentrate in specific epochs of follow-up [22]. Oxygenation markers (average/min SpO_2_; time < 90%) showed closer correspondence with dynamic HRs than with three time frames. That is expected: desaturation burden is a continuous, cumulative exposure whose relationship to outcomes is non-linear and time-dependent. Dynamic models capture this dose–time relation; binning underestimates it. Prior studies identify hypoxic burden as a dominant driver of cardiovascular morbidity/mortality in OSA [22]. Mixed or inverse associations for hypopneas and NREM-dominant events are also easier to interpret dynamically: they can drift below 1.0 in specific periods, reflecting lower pathophysiologic load or competing risks, whereas interval HRs obscure such temporal heterogeneity by enforcing average effects per bin.

Despite the robust findings, this study has certain limitations. Being a single-center cohort study, it is subject to inherent biases related to data collection and patient selection. The single-center nature of the study may also limit the generalizability of the results. Future research should aim to validate these findings in larger, multi-center cohorts and explore the integration of other emerging biomarkers and technological advancements in the assessment of OSA. The exclusion of patients with total sleep times below 150 min or missing variables may introduce selection bias, limiting generalizability. Residual confounding factors may remain, particularly for complex interactions between OSA and its comorbidities. At the stage of data collection, standardized criteria for central hypopneas had not yet been published (2025) [9]. Therefore, as the authors of this study, we do not specifically address these respiratory events

## 5. Conclusions

The highest long-term mortality risk was associated with severe OSA, particularly events occurring during REM sleep and significant oxygen desaturation. Dynamic modeling revealed temporal patterns of risk progression, providing insights beyond those captured by static, point-in-time analyses.

AHI_REM_ HR with 95% CI dynamic curve was related to time with desaturation below 90% and AHI_NREM_ with minimal SpO_2_, while the AHI_TST_ curve corresponded with average SpO_2_.

## Figures and Tables

**Figure 1 arm-93-00046-f001:**
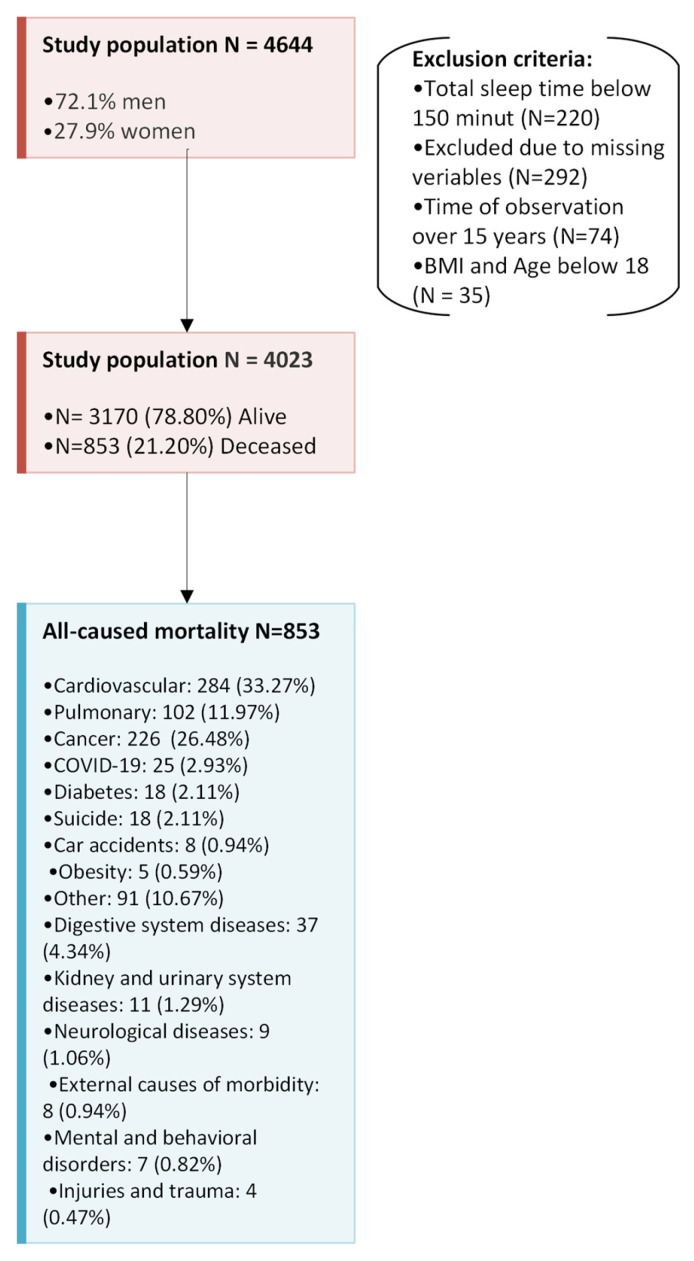
A schematic overview of the study design, including the applied exclusion criteria and the classification of causes of death according to the ICD-10 codes.

**Figure 2 arm-93-00046-f002:**
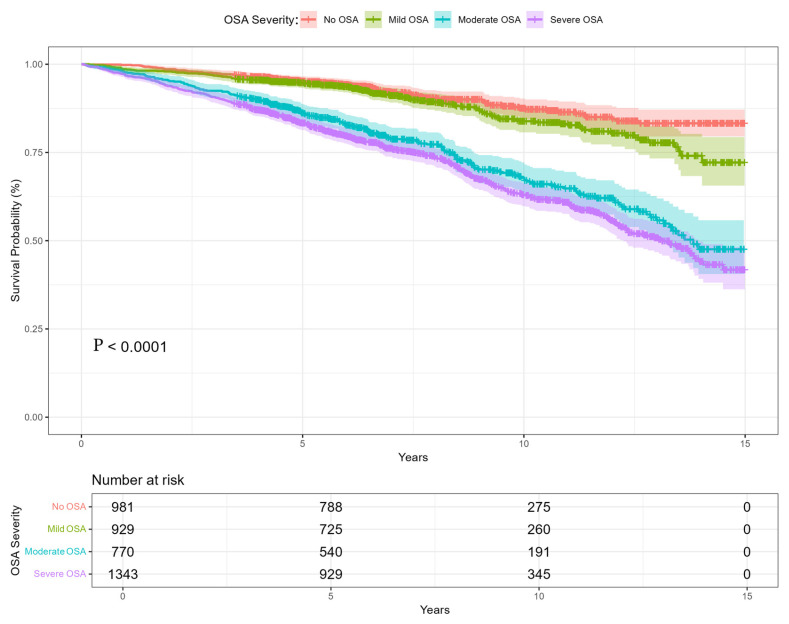
The Kaplan–Meier survival curve from 0–15 years of observations. Separate curves related to OSA severity: No OSA, mild OSA, moderate OSA, and severe OSA.

**Figure 3 arm-93-00046-f003:**
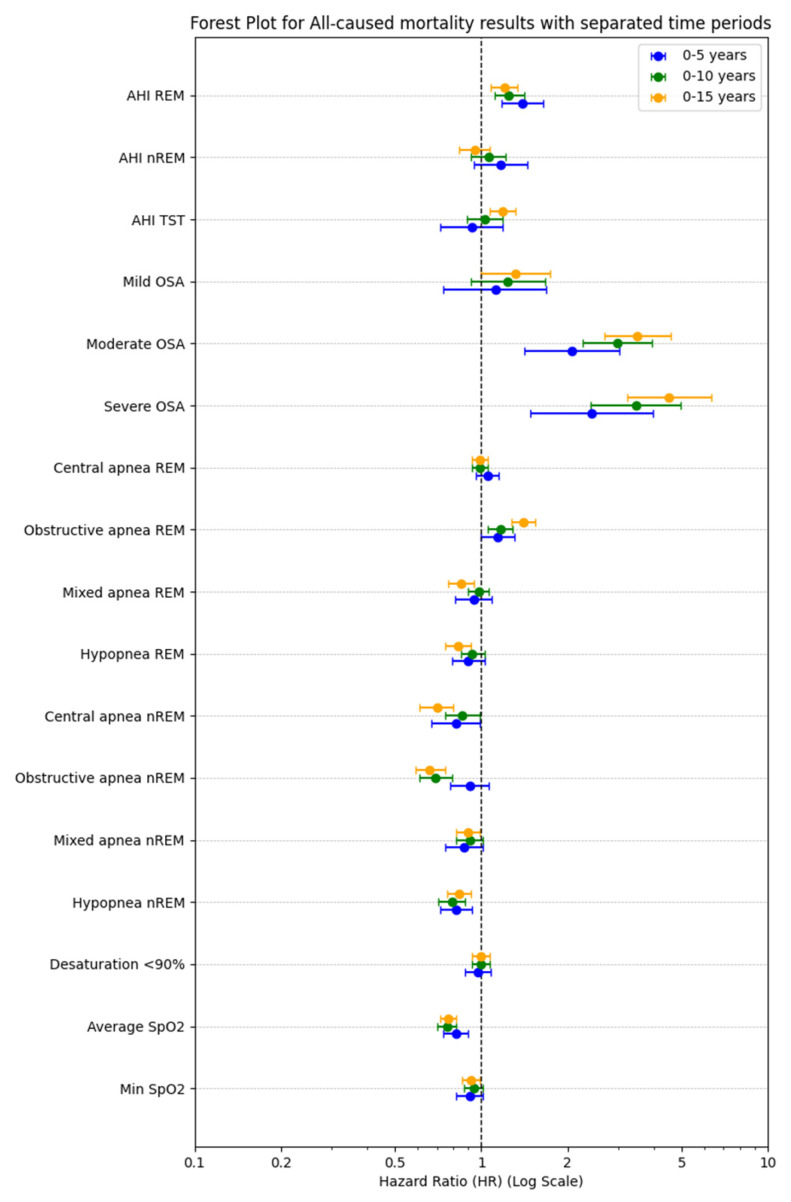
Polysomnographic variables of interest across mortality in the three observation periods (0–5 years, 0–10 years, 0–15 years). Each point represents the effect size (odds ratio) and confidence intervals according to multivariable ROC analysis.

**Figure 4 arm-93-00046-f004:**
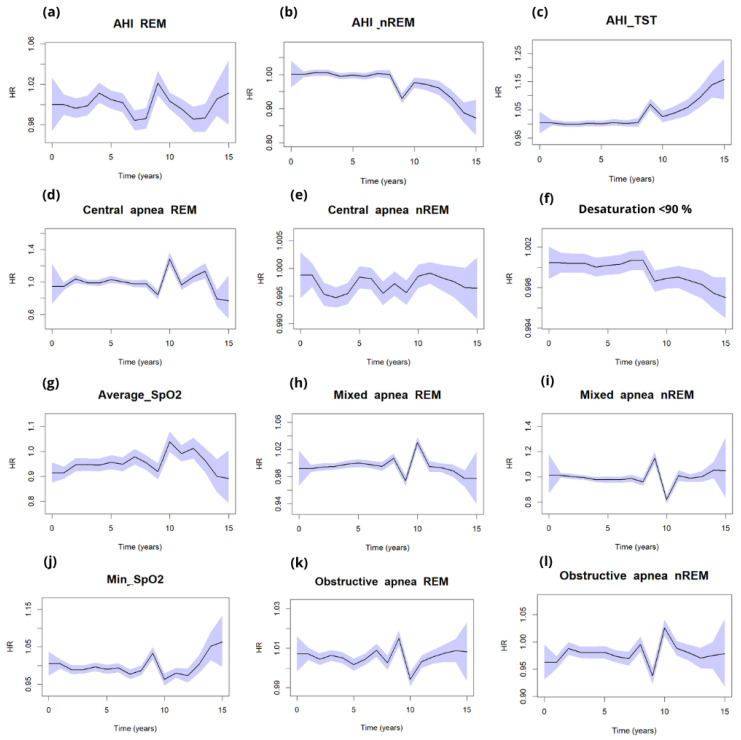
Dynamic changes of HR with light blue 0.95 CI in timeframe from 0–15 years in all-cause mortality for (**a**) AHI_REM_, (**b**) AHI_NREM_, (**c**) AHI_TST_, and respiratory events: REM and NREM apneas ((**k**,**l**) obstructive, (**d**,**e**) central, (**h**,**i**) mixed), (**g**) average desaturations, (**f**) desaturation below 90% saturations, and (**j**) minimal saturations.

**Table 1 arm-93-00046-t001:** Baseline characteristics of the overall study population *n* = 4023.

	*n* = 4023				
Variables	*n*	Mean	SD	Min	Max
Age	4023	55.88	12.88	19	93
BMI ‡	4023	31.69	6.01	18.08	59.97
Neck circumference	4023	42.61	3.55	30	60
TST	4023	336.06	70.19	152	529.3
Wake	4023	54.52	49.58	0	330
REM	4023	60.76	30.79	0	278
NREM	4023	278.47	60.06	0	481
AHI_REM_ ¶	4023	18.49	25.38	0	143.3
AHI_NREM_ ¶	4023	18.39	26.77	0	158.5
AHI_TST_ ¶	4023	26.68	27.58	0	294
Average saturation	4023	91.55	3.78	49	98
Time < 90% saturation	4023	76.94	14.44	0.7	98
BP systolic ~	4023	135.9	18.2	90	250
BP diastolic ~	4023	86.09	11.56	50	140
Epworth SS ‖	4023	8.35	4.69	0	24
Morning fatigue †	2986 (74.22%)				
Snoring †	3631 (90.26%)				
Morning headache †	1736 (43.15%)				
Hypertension †	2459 (61.12%)				
Atrial fibrillation †	202 (5.02%)				
Dyslipidemia †	953 (23.69%)				
Stroke †	163 (4.05%)				
Myocardial Infraction †	202 (5.02%)				
Depression †	186 (4.62%)				
Diabetes †	654 (16.26%)				
Hypothyroidism †	161 (4.00%)				
Post thyroidectomy †	75 (1.86%)				
Hyperthyroidism †	38 (0.94%)				
Oral hypoglycemic agents †	496 (12.33%)				
Insulin †	126 (3.13%)				
ASA †	722 (17.95%)				
Vitamin K antagonist	100 (2.49%)				
New oral anticoagulants	75 (1.86%)				
No OSA (AHI < 5) ¶	981 (24.38%)				
Mild OSA (AHI ≥ 5. AHI < 15) ¶	929 (23.09%)				
moderate OSA (AHI ≥ 15. AHI < 30) ¶	770 (19.14%)				
Severe OSA (AHI ≥ 30) ¶	1343 (33.38%)				

*n*—number of observations; values are means with SD. Average saturation and time with saturation < 90 denotes oxygen saturation level as measured by pulse oximetry. TST—total sleep time [minutes]; Wake—total wake recorded [minutes]; REM—rapid eye movement phase of sleep [minutes] NREM—non-rapid eye movement phase of sleep [minutes]. † Symptoms, comorbidities, and drug usage were self-reported. ‡ The body mass index is the weight in kilograms divided by the square of the height in meters. ¶ The apnea–hypopnea index (AHI) is the number of apnea and hypopnea episodes per hours of sleep time. ‖ Scores on the Epworth Sleepiness Scale range from 0.0 to 24.0, with higher scores indicating more daytime sleepiness. ~ Blood pressure (BP) was measured twice by the physician before polysomnography, including systolic and diastolic readings, as well as the delta difference between systolic and diastolic values.

**Table 2 arm-93-00046-t002:** Characteristics of causes of death, number of patients, gender, OS year mean with SD, and AHI_REM_, AHI_NREM_, and AHI_TST_ means with SD.

	Specific Cause of Death	*n* (%)	Gender M/F	OS TIME YEARS Mean (SD)	AHI_REM_ Mean (SD)	AHI_NREM_ Mean (SD)	AHI_TST_ Mean (SD)
Cardiovascular	Chronic heart failure	97 (11.37%)	82/15	5.33 (3.14)	31.05 (28.67)	31.70 (28.70)	39.57 (30.27)
	Myocardial infarction	72 (8.44%)	56/16	4.67 (3.39)	31.93 (32.58)	36.04 (36.71)	44.91 (45.26)
	Stroke, intracerebral hemorrhage	62 (7.27%)	43/19	5.45 (3.54)	31.99 (29.86)	29.92 (32.49)	38.85 (30.40)
	Arteriosclerosis	17 (1.99%)	14/3	6.48 (4.22)	31.77 (27.14)	32.06 (24.26)	38.10 (28.36)
	Cardiomyopathy	11 (1.29%)	9/2	5.42 (3.89)	38.55 (37.99)	36.90 (33.75)	43.19 (29.61)
	Aneurysm	11 (1.29%)	10/1	5.98 (3.63)	32.61 (27.94)	37.00 (32.36)	42.71 (26.08)
	Cardiac arrest	6 (0.70%)	5/1	4.63 (3.81)	18.87 (18.07)	28.08 (44.86)	33.95 (42.04)
	Endocarditis	6 (0.70%)	1/1	4.03 (2.62)	23.82 (26.09)	23.12 (36.78)	30.70 (28.87)
	Arterial hypertension	2 (0.23%)	42/10	2.82 (2.70)	49.60 (42.99)	59.60 (54.02)	57.90 (52.61)
Pulmonary	Chronic obstructive pulmonary disease	52 (6.10%)	25/9	4.81 (3.13)	35.38 (30.12)	37.70 (37.27)	49.63 (34.80)
	Pneumonia	34 (3.99%)	9/3	7.89 (4.27)	31.86 (28.01)	34.37 (33.10)	36.24 (30.64)
	Pulmonary embolism	12 (1.41%)	2/1	5.42 (3.74)	30.72 (35.37)	37.89 (38.69)	56.08 (33.63)
	Interstitial pulmonary disease	3 (0.35%)	1/0	4.54 (5.36)	31.00 (53.69)	51.03 (74.08)	55.53 (64.07)
	Respiratory failure	1 (0.12%)	1/0	3.55 (nan)	72.90 (nan)	20.20 (nan)	27.40 (nan)
COVID-19		25 (2.93%)	22/3	6.89 (3.29)	27.47 (25.05)	30.32 (26.36)	37.10 (22.80)
Diabetes		18 (2.11%)	13/5	5.81 (3.69)	25.55 (31.36)	35.11 (34.82)	47.81 (25.06)
Suicide		18 (2.11%)	14/4	3.99 (2.62)	19.73 (25.55)	16.80 (24.45)	17.54 (24.15)
Car accidents		8 (0.94%)	7/1	4.31 (3.68)	14.41 (16.00)	17.84 (22.57)	19.58 (19.10)
Obesity		5 (0.59%)	4/1	4.34 (3.77)	61.98 (20.19)	59.22 (33.46)	60.02 (30.83)
Cancer	Lung cancer	69 (8.09%)	52/17	5.12 (3.43)	30.83 (28.65)	31.82 (33.59)	38.69 (31.24)
	Gastrointestinal cancer	61 (7.15%)	43/18	5.85 (3.64)	31.70 (27.35)	29.25 (28.54)	35.41 (25.97)
	Genitourinary and reproductive system cancer	22 (2.58%)	16/6	7.03 (3.34)	32.63 (36.48)	39.21 (37.26)	44.50 (34.04)
	Hematopoietic and lymphatic system neoplasms	20 (2.34%)	11/9	6.67 (3.13)	34.80 (30.17)	32.60 (32.21)	33.59 (30.55)
	Skin cancer	16 (1.88%)	11/5	6.38 (3.61)	11.99 (14.76)	11.06 (16.27)	23.16 (27.39)
	Unspecified or metastatic neoplasms	13 (1.52%)	12/1	6.58 (3.57)	38.94 (36.48)	33.44 (22.39)	34.97 (23.33)
	Nervous system neoplasms	11 (1.29%)	6/5	5.89 (2.80)	33.73 (28.11)	35.66 (33.87)	40.67 (28.90)
	Breast cancer	6 (0.70%)	0/6	6.85 (3.45)	25.13 (21.31)	24.72 (28.55)	26.18 (26.87)
	Head and neck cancer	6 (0.70%)	6/0	7.53 (3.33)	30.40 (26.51)	19.70 (34.57)	22.83 (32.34)
	Endocrine neoplasms	2 (0.23%)	1/1	7.25 (3.30)	25.10 (35.50)	20.40 (28.85)	26.45 (21.85)
Other		91 (10.67%)	78/13	5.71 (4.09)	32.25 (30.69)	38.57 (37.38)	46.28 (34.89)
Digestive system diseases		37 (4.34%)	27/10	6.53 (3.79)	25.94 (27.27)	26.96 (27.04)	37.60 (28.31)
Kidney and urinary system diseases		11 (1.29%)	6/3	5.78 (2.96)	21.10 (28.97)	27.04 (33.18)	34.06 (28.94)
Neurological diseases		9 (1.06%)	6/2	4.72 (2.69)	29.78 (25.47)	26.54 (26.04)	26.98 (22.95)
External causes of morbidity		8 (0.94%)	5/2	7.23 (4.16)	25.85 (28.48)	30.65 (25.19)	30.41 (25.15)
Mental and behavioral disorders		7 (0.82%)	0/6	6.44 (3.87)	29.27 (25.62)	30.20 (29.29)	35.53 (25.10)
Injuries and trauma		4 (0.47%)	3/1	4.66 (3.49)	40.77 (32.35)	57.00 (44.60)	56.30 (37.86)

**Table 3 arm-93-00046-t003:** Multivariable analysis for each timeframe expressed separately with adjusted hazard ratios, 95% confidence intervals, and *p*-values. Variables that became non-significant after Bonferroni correction are indicated with an asterisk (*).

Variables	0–5 Years HR (Multivariable)	0–10 Years HR (Multivariable)	0–15 Years HR (Multivariable)
Age	2.16 (1.99–2.46, *p* < 0.001)	2.26 (2.09–2.45, *p* < 0.001)	2.34 (2.18–2.54, *p* < 0.001)
Body mass index (BMI)	1.04 (0.92–1.18, *p* = 0.507)	1.18 (1.08–1.29, *p* < 0.001)	1.20 (1.10–1.30, *p* < 0.001)
Epworth Sleepiness Scale	1.28 (1.16–1.42, *p* < 0.001)	1.20 (1.11–1.30, *p* < 0.001)	1.17 (1.09–1.26, *p* < 0.001)
Neck circumference	1.21 (1.07–1.37, *p* = 0.002)	1.12 (1.02–1.22, *p* = 0.013) *	1.15 (1.06–1.24, *p* = 0.001)
Systolic blood pressure	1.02 (1.01–1.02, *p* < 0.001)	1.02 (1.02–1.02, *p* < 0.001)	1.02 (1.02–1.02, *p* < 0.001)
Diastolic blood pressure	0.98 (0.97–1.00, *p* = 0.004)	0.97 (0.96–0.98, *p* < 0.001)	0.96 (0.96–0.97, *p* < 0.001)
Morning fatigue	0.78 (0.62–0.98, *p* = 0.031) *	0.71 (0.60–0.84, *p* < 0.001)	0.90 (0.77–1.05, *p* = 0.185)
Sleepiness	0.88 (0.63–1.22, *p* = 0.436)	0.92 (0.71–1.19, *p* = 0.518)	0.81 (0.64–1.03, *p* = 0.081)
Snoring	0.96 (0.70–1.33, *p* = 0.823)	0.76 (0.60–0.96, *p* = 0.023) *	0.95 (0.75–1.19, *p* = 0.639)
Morning headaches	0.96 (0.79–1.18, *p* = 0.709)	1.01 (0.87–1.18, *p* = 0.886)	0.94 (0.82–1.08, *p* = 0.362)
Hypertension	1.57 (1.23–2.00, *p* < 0.001)	1.66 (1.39–1.99, *p* < 0.001)	1.72 (1.45–2.03, *p* < 0.001)
Diabetes	1.40 (0.95–2.07, *p* = 0.092)	1.30 (0.99–1.71, *p* = 0.056)	1.38 (1.07–1.78, *p* = 0.013) *
Atrial fibrillation	1.18 (0.79–1.77, *p* = 0.425)	1.05 (0.78–1.42, *p* = 0.752)	1.33 (1.00–1.77, *p* = 0.053)
Dyslipidemia	0.91 (0.72–1.14, *p* = 0.411)	0.86 (0.72–1.02, *p* = 0.086)	0.96 (0.82–1.13, *p* = 0.639)
Depression	0.65 (0.35–1.18, *p* = 0.157)	0.62 (0.41–0.94, *p* = 0.023)	0.66 (0.45–0.96, *p* = 0.031) *
Stroke	2.41 (1.66–3.49, *p* < 0.001)	1.53 (1.17–1.99, *p* = 0.002)	1.77 (1.38–2.28, *p* < 0.001)
Myocardial Infarction	0.29 (0.14–0.62, *p* = 0.001)	0.39 (0.22–0.68, *p* = 0.001)	0.40 (0.23–0.68, *p* = 0.001)
Hypothyroidism	1.34 (0.78–2.29, *p* = 0.291)	2.37 (1.50–3.75, *p* < 0.001)	1.81 (1.18–2.76, *p* = 0.006)
Post thyroidectomy	2.00 (0.88–4.52, *p* = 0.096)	1.78 (0.92–3.45, *p* = 0.087)	1.34 (0.72–2.51, *p* = 0.353)
Hyperthyroidism	0.79 (0.52–1.18, *p* = 0.251)	1.17 (0.88–1.54, *p* = 0.278)	1.26 (0.97–1.65, *p* = 0.084)
Oral hypoglycemic agents	1.71 (1.14–2.58, *p* = 0.010) *	1.42 (1.04–1.95, *p* = 0.028)	1.64 (1.23–2.19, *p* = 0.001)
Insulin	2.29 (1.06–4.93, *p* = 0.034) *	1.90 (0.97–3.73, *p* = 0.063)	1.44 (0.74–2.80, *p* = 0.288)
Vitamin K antagonist	1.40 (0.77–2.54, *p* = 0.269)	1.78 (1.14–2.78, *p* = 0.011)	1.80 (1.17–2.76, *p* = 0.007)
New oral anticoagulants	2.56 (1.67–3.94, *p* < 0.001)	2.07 (1.44–2.98, *p* < 0.001)	1.71 (1.21–2.43, *p* = 0.002)
Aspirin (ASA)	1.56 (1.24–1.96, *p* < 0.001)	1.58 (1.32–1.88, *p* < 0.001)	1.46 (1.24–1.72, *p* < 0.001)
AHI_REM_	1.38 (1.18–1.62, *p* < 0.001)	1.26 (1.12–1.41, *p* < 0.001)	1.23 (1.11–1.37, *p* < 0.001)
AHI_NREM_	1.14 (0.92–1.41, *p* = 0.222)	1.10 (0.96–1.26, *p* = 0.165)	1.02 (0.91–1.15, *p* = 0.684)
AHI_TST_	1.09 (0.85–1.38, *p* = 0.502)	1.12 (1.00–1.25, *p* = 0.047) *	1.27 (1.17–1.37, *p* < 0.001)
Mild OSA	1.14 (0.75–1.72, *p* = 0.538)	1.34 (1.00–1.80, *p* = 0.049) *	1.39 (1.06–1.84, *p* = 0.018) *
Moderate OSA	2.10 (1.43–3.08, *p* < 0.001)	3.10 (2.35–4.09, *p* < 0.001)	3.65 (2.80–4.74, *p* < 0.001)
Severe OSA	2.34 (1.43–3.82, *p* = 0.001)	3.74 (2.64–5.28, *p* < 0.001)	4.72 (3.40–6.56, *p* < 0.001)
Central apnea REM	1.03 (0.94–1.13, *p* = 0.511)	0.97 (0.91–1.04, *p* = 0.360)	0.97 (0.91–1.03, *p* = 0.286)
Obstructive apnea REM	1.14 (1.00–1.31, *p* = 0.049) *	1.18 (1.06–1.31, *p* = 0.002)	1.38 (1.26–1.51, *p* < 0.001)
Mixed apnea REM	0.96 (0.83–1.10, *p* = 0.549)	0.96 (0.88–1.05, *p* = 0.408)	0.87 (0.79–0.96, *p* = 0.006) *
Hypopnea REM	0.92 (0.81–1.05, *p* = 0.219)	0.91 (0.82–1.00, *p* = 0.049) *	0.81 (0.73–0.89, *p* < 0.001)
Central apnea NREM	0.82 (0.67–1.00, *p* = 0.045) *	0.86 (0.75–1.00, *p* = 0.045) *	0.70 (0.61–0.79, *p* < 0.001)
Obstructive apnea NREM	0.90 (0.77–1.06, *p* = 0.201)	0.70 (0.62–0.80, *p* < 0.001)	0.67 (0.60–0.75, *p* < 0.001)
Mixed apnea NREM	0.86 (0.74–1.00, *p* = 0.046) *	0.93 (0.84–1.02, *p* = 0.133)	0.91 (0.83–1.01, *p* = 0.066)
Hypopnea NREM	0.80 (0.71–0.91, *p* = 0.001)	0.83 (0.75–0.93, *p* = 0.001)	0.88 (0.80–0.97, *p* = 0.008) *
Desaturation < 90%ent	1.15 (1.05–1.25, *p* = 0.002)	1.06 (1.00–1.13, *p* = 0.064)	1.06 (1.00–1.13, *p* = 0.044)
Average SpO_2_	0.84 (0.76–0.93, *p* < 0.001)	0.78 (0.73–0.83, *p* < 0.001)	0.82 (0.78–0.87, *p* < 0.001)
Min SpO_2_	0.86 (0.79–0.94, *p* = 0.001)	0.85 (0.79–0.91, *p* < 0.001)	0.82 (0.78–0.87, *p* < 0.001)

## Data Availability

The database may be shared only upon the explicit agreement of the first and corresponding author of the manuscript.

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
