# Peer review of "Beyond the Apnea–Hypopnea Index: Exploring Time-Dependent Hazard Ratios of Respiratory Events in Obstructive Sleep Apnea"

_arm, 2025, doi:10.3390/arm93050046_

Round 1

Reviewer 1 Report

Comments and Suggestions for Authors

Dear Authors

It is interesting for long term follow up and association of PSG variables.

Regarding hypoxic burden and heart rate variability and prognosis in patient with OSA ,in future study comparison of all parameters must be included.

Author Response

Comments 1:

Regarding hypoxic burden and heart rate variability and prognosis in patient with OSA ,in future study comparison of all parameters must be included.

Response 1: 

On behalf of myself and the co-authors, we would like to thank you for your review. We agree with the Reviewer’s comment. As soon as the software provider enables the calculation of hypoxic burden as well as the respiratory arousal threshold, we will conduct the appropriate post-hoc analysis and publish the results.

Reviewer 2 Report

Comments and Suggestions for Authors

This retrospective research includes more than 4000 patients referred for suspected sleep apnea with up 15 years of follow up . The purpose of the study is to identify predictors of mortality in OSA patients. The main novelty lies in the application of dynamic hazard ratio modelling, that reflectsthe evolving nature of OSA-related risks. Another strength of this research is the long follow-up of up to 15 years and the large sample size. The authors conclude that respiratory events during Rem-sleep and nocturnal hypoxemia are stronger predictors of mortality than the traditional criteria of apnea-hypopnea index (AHI).

There are some limitations that I would like authors to explain;

  1. Inclusion and exclusion criteria: age at entry into the study seems not having been taken into account . It is not the same risk of comorbidities or death at 30 years of age than at 80 years of age...   may be there are more cofounders that can affect results . 
  2. It is not mentioned if any treatment as CPAP, surgery , advancement devices,... was employed in more than 4000 patients followed up to 15 years. It is surprising that ethic comitte accept so many apnea patients without treatment so much time... all of them refused CPAP or any other treatment ? Treatment would affect  risks associated with OSA. 
  3. Apparently "protective" effect of depression are likely due to residual confounding factors and require further investigation
  4. In discussion section authors could add similar long-term studies comparing findings-

I consider this manuscript , despite mentioned limitations, makes significant contribution demonstrating that Rem events and nocturnal hypoxemia are stronger predictors of mortality than AHI, but I ask authors to answer my questions.

Author Response

Comments 1:

Inclusion and exclusion criteria: age at entry into the study seems not having been taken into account . It is not the same risk of comorbidities or death at 30 years of age than at 80 years of age...   may be there are more cofounders that can affect results . 

Response 1: 

We greatly appreciate the Reviewer’s insightful comment. However, the age of the patients included in our study does not raise concerns. Age is indeed a risk factor, which was confirmed in our additional univariable analysis (not included in the manuscript). Nevertheless, when assessing other clinical variables in the multivariable analysis, this effect was not observed. Our findings appear to be consistent with the study by Azarian M, Ramezani A, Sharafkhaneh A, et al. The Association between All-cause Mortality and Obstructive Sleep Apnea in Adults: A U-Shaped Curve. Ann Am Thorac Soc. 2025, in which the authors emphasized both the variability of mortality risk with age and the atypical U-shaped curve.

Comments 2:

It is not mentioned if any treatment as CPAP, surgery , advancement devices,... was employed in more than 4000 patients followed up to 15 years. It is surprising that ethic comitte accept so many apnea patients without treatment so much time... all of them refused CPAP or any other treatment ? Treatment would affect  risks associated with OSA. 

Response 2: 

We added, line 145-150: 

All patients who received the results of the polysomnographic examination were given detailed recommendations regarding the need to initiate treatment in accordance with current guidelines: weight reduction, mandibular advancement devices (MAD), positional therapy, and CPAP therapy.

Furthermore, it should be emphasized that the present study does not aim to evaluate the impact of therapy, the variety of therapeutic interventions, or treatment compliance. Measures of therapeutic efficacy and effectiveness are beyond the scope of this analysis. The potential influence of therapy is assumed to be identical for both the hazard ratio and the dynamic hazard ratio; however, in the authors’ view, the differences observed between these measures are clinically meaningful.

Comments 3:

Apparently "protective" effect of depression are likely due to residual confounding factors and require further investigation

Response 3: 

Line 265-270: "The prevalence of depression and suicides (even higher than car accidents) among patients referred to sleep disorder clinics reflects the multifaceted nature of their health issues. Notably, depression was associated with a reduced risk of all-cause mortality (HR 0.66, 95% CI: 0.45–0.96, p=0.031), underscoring the necessity of diagnostics and treatment within interdisciplinary clinical teams." 

These findings warrant further investigation. It may be hypothesized that psychiatric patients demonstrate an increased sensitivity to specific signs and symptoms, which subsequently leads to more frequent consultation with healthcare providers.

Comments 4:

In discussion section authors could add similar long-term studies comparing findings-

Response 4: 

We have included the following paragraph; nevertheless, the scarcity of studies in the literature addressing the dynamic hazard ratio represents a significant limitation. This paucity of evidence makes it challenging for the authors to place the present findings in the context of existing research and to draw direct comparisons with previously reported results.

To explore the topic of dynamic hazard ratios (HRs) in obstructive sleep apnea (OSA), particularly in relation to the variations of risk factors over time and across different clinical contexts, several pertinent studies have been identified. These studies illustrate the significance of utilizing dynamic modeling approaches to better understand the impact of OSA on mortality and other health-related outcomes. For instance, [16]Kim et al. Kim et al. (2020) examined various polysomnographic characteristics associated with OSA and their implications on mortality [16]They noted that metrics such as the ratio of REM to NREM apneas could indicate physiological differences that might influence hazard ratios over time, emphasizing the importance of understanding how different patterns of sleep disturbances correlate with long-term outcomes [16,17]. 

Reviewer 3 Report

Comments and Suggestions for Authors

MAJOR REVISIONS
1. Methodological Concerns
Dynamic Hazard Ratio Analysis
•    The justification for dynamic HR modeling (Discussion, lines 300-326) appears post-hoc. The Methods section should clearly state the a priori rationale for this approach and describe the statistical methodology in detail (e.g., spline specifications, knot placement, smoothing parameters).
•    Figures 4-17 lack adequate legends and axis labels. Each figure needs clear descriptions of what is being displayed, confidence interval calculations, and sample sizes at different time points.
Statistical Analysis Issues
•    Multiple testing correction is absent despite analyzing numerous variables across three time periods. Consider false discovery rate adjustment or Bonferroni correction.
•    The exclusion of 292 patients with "crucial variables missing" needs specification of which variables and potential selection bias discussion.
•    Cox proportional hazards assumption testing is not mentioned. This is critical given the 15-year follow-up period.
2. Results Presentation
Table Organization
•    Table 3 is overwhelming with 40+ variables. Consider separating into supplementary tables by category or creating a focused main table with key findings.
•    The claim "AHIREM HR with 95% CI dynamic curve was related to time with desaturation below 90%" (lines 241-242) lacks statistical support. Provide correlation coefficients or formal interaction tests.
Missing Critical Information
•    CPAP treatment data is completely absent. This is a major confounder for OSA mortality studies.
•    No information on OSA treatment adherence or follow-up sleep studies.
•    Cardiovascular event timing relative to PSG is not provided.
3. Clinical Context
Study Population
•    The 33.4% severe OSA prevalence seems high for a referred population. Discuss potential referral bias.
•    BMI range (18.08-59.97) is extreme. Consider sensitivity analyses excluding extreme values.
•    The control group (AHI<5) may include symptomatic patients who don't meet OSA criteria. This needs clarification.
MINOR REVISIONS
1. Writing and Clarity
Abstract
•    Line 37: "Single-center study conducted..." needs grammatical revision.
•    Lines 44-45: Simplify the complex sentence about dynamic hazard ratios.
Introduction
•    Lines 73-81: The paragraph on AHI subtypes disrupts flow. Consider moving to Methods.
•    Line 89: Add specific hypotheses rather than general "diverse predictors."
2. Technical Corrections
Methods
•    Line 94: "8:00 pm" should be "20:00" for consistency.
•    Lines 106-114: The definitions of comorbidities need standardization (ICD codes preferred).
•    Line 150: "attained" should be "defined as."
Tables and Figures
•    Table 1: "sie.35" appears to be a typo (should be ~8.35 based on context).
•    Table 2: "OS_TIME_YEARS" header needs clarification.
•    Figure 1: The flowchart is referenced but appears to be missing or mislabeled.
3. Statistical Reporting
•    Line 165: Specify R package versions for reproducibility.
•    Lines 170-171: The log-rank test is mentioned for survival curves but results aren't clearly reported.
•    Confidence intervals should be consistently formatted (e.g., "95% CI: 1.18-1.62" throughout).
4. Discussion Points
Interpretation Issues
•    Lines 261-264: The depression finding (protective effect) needs more cautious interpretation. Could this reflect survivorship bias or treatment effects?
•    Lines 275-284: The differential effects of AHIREM vs AHINREM need physiological context earlier in the manuscript.

Author Response

MAJOR REVISIONS

Dynamic Hazard Ratio Analysis
•    The justification for dynamic HR modeling (Discussion, lines 300-326) appears post-hoc. The Methods section should clearly state the a priori rationale for this approach and describe the statistical methodology in detail (e.g., spline specifications, knot placement, smoothing parameters).

Response: We decided to include dynamic hazard models as a complementary analysis to better illustrate the potential changes in hazard ratios over time. In the Methods section, we added details regarding the package employed and the analytical approach.
“In addition to the standard Cox proportional hazards model, we applied a dynamic hazard model implemented in the R package dynamichazard (function ddhazard) to explore and visualize potential time‐dependent effects. This approach treats the regression coefficients as a time‐evolving state vector following a random‐walk process, while the baseline hazard is approximated by a piecewise constant function over a predefined time grid.”

  •    Figures 4-17 lack adequate legends and axis labels. Each figure needs clear descriptions of what is being displayed, confidence interval calculations, and sample sizes at different time points.
    Statistical Analysis Issues

Response: We have corrected the figures according to the reviewer's suggestion.
•    Multiple testing correction is absent despite analyzing numerous variables across three time periods. Consider false discovery rate adjustment or Bonferroni correction.

Response: We applied Bonferroni correction for multiple testing. Variables that lost statistical significance after this adjustment are marked with an asterisk (*) next to the adjusted p-value in both the univariable and multivariable analysis tables.
•    The exclusion of 292 patients with "crucial variables missing" needs specification of which variables and potential selection bias discussion.

Response: A total of 292 patients were excluded from the analysis due to ‘crucial variables missing’. This group included patients with missing information on reported clinical symptoms at admission, missing data on comorbidities or medication use, incorrectly recorded anthropometric data (e.g., height 66 cm with weight 172 kg due to data transposition), as well as duplicated patients caused by repeated polysomnographic assessment during follow-up. In the authors’ opinion, the careful exclusion of 292 patients (7.2% of the total cohort) would not have a significant impact on the presented results 

  •    Cox proportional hazards assumption testing is not mentioned. This is critical given the 15-year follow-up period.

Response: Although the Schoenfeld residuals test did not indicate significant violations of the proportional hazards assumption (all p > 0.05). We mention the Schoenfeld residuals in the methodology section. 

We have added the following fragment to the Methods section:

“The proportional hazards (PH) assumption was evaluated for each covariate using Schoenfeld residuals with the cox.zph function from the survival package. The global and variable–specific tests did not indicate significant violations (all p > 0.05). In addition to the standard Cox proportional hazards model, we applied a dynamic hazard model implemented in the R package dynamichazard (function ddhazard) to explore and visualize potential time‐dependent effects. This approach treats the regression coefficients as a time‐evolving state vector following a random‐walk process, while the baseline hazard is approximated by a piecewise constant function over a predefined time grid.”

  1. Results Presentation
    Table Organization
    •    Table 3 is overwhelming with 40+ variables. Consider separating into supplementary tables by category or creating a focused main table with key findings.

Response: We cannot agree with the reviewer’s comment. According to the adopted statistical analysis model, in which the hazard ratio in the multivariate analysis was assessed for three groups of interrelated factors (lines 174–177: ‘To perform multivariate modelling, we separated all study variables into three groups: (1) clinical signs and symptoms, measurements such as weight, height, BMI, blood pressure; (2) comorbidities and medications; (3) polysomnographic variables, see Table 3’). Excluding variables from the table would mislead the reader, suggesting that the presented associations were observed only for a selected subset of factors, rather than for all variables included in the analysis. Moreover, in the Appendix we provide a separate table presenting the univariate analysis, where each variable was analyzed individually.
•    The claim "AHIREM HR with 95% CI dynamic curve was related to time with desaturation below 90%" (lines 241-242) lacks statistical support. Provide correlation coefficients or formal interaction tests.
Missing Critical Information

Response: We removed minor text to improve the clarity.
•    CPAP treatment data is completely absent. This is a major confounder for OSA mortality studies.
•    No information on OSA treatment adherence or follow-up sleep studies.
•    Cardiovascular event timing relative to PSG is not provided.

Response: We acknowledge the reviewer’s comment. The omission of PAP therapy data was a conscious decision by the authors. Our rationale is as follows:

In Poland, the choice of PAP device is typically made by the patient. There are at least six manufacturers on the market, each offering at least four different device types. The algorithms for detecting respiratory events vary significantly between devices (e.g., Philips vs. ResMed), which would introduce data bias. Additionally, over the 15-year observation period, monitoring methods for PAP therapy have changed—from in-person visits to remote monitoring.

Publications in reputable journals such as NEJM, AJRCCM, and JAMA acknowledge the effectiveness of PAP therapy in reducing mortality when used consistently on a daily basis. However, global experts emphasize that the effect of PAP therapy is difficult to isolate due to other modifiable factors, such as weight loss (e.g., McEvoy et al. NEJM, Chirinos et al. NEJM, Jennum Nature and Science of Sleep).

Furthermore, unlike many clinical trials, the patients included in our analysis were not highly selected, which is typically the case in studies focusing solely on PAP therapy effectiveness. In the recently published paper by A. Azarbarzin et al. in the European Heart Journal, “Cardiovascular benefit of continuous positive airway pressure according to high-risk obstructive sleep apnoea: a multi-trial analysis” on 3,549 participants, the authors concluded: “Continuous positive airway pressure preferentially improves cardiovascular outcomes in high-risk OSA, while harm in low-risk OSA may counteract this effect.”

Given all the aforementioned arguments, we would like to emphasize that the absence of presented results on PAP therapy or other forms of therapy has the same impact on the hazard ratio and dynamic hazard ratio, while the differences in hazards of the analyzed variables form the basis of the study, rather than the assessment of the impact of PAP therapy on mortality.

  1. Clinical Context
    Study Population
    •    The 33.4% severe OSA prevalence seems high for a referred population. Discuss potential referral bias.

In the region of Łódź, Poland, between 2005 and 2015, there was only one sleep center equipped with four PSG devices, serving a population of 2.3 million inhabitants. The higher prevalence of severe OSA may have resulted from pre-diagnostic screening of severe cases in outpatient clinics, which referred patients for diagnostic evaluation in the sleep department.

  •    BMI range (18.08-59.97) is extreme. Consider sensitivity analyses excluding extreme values.

Response: We would prefare to keep this range of BMI.
   The control group (AHI<5) may include symptomatic patients who don't meet OSA criteria. This needs clarification.

The study group of 4023 patients was included in the statistical analysis. Among them, 982 patients (24.4%) were referred with a presumptive diagnosis of OSA based on typical symptoms but did not meet the diagnostic criteria for OSA (AHI <5) and served as the control group.

MINOR REVISIONS

  1. Writing and Clarity
    Abstract
    •    Line 37: "Single-center study conducted..." needs grammatical revision.

Response: “A single-center study was conducted”
•    Lines 44-45: Simplify the complex sentence about dynamic hazard ratios.

Response: “The hazard ratio analysis showed that mortality risk changed over time depending on sleep stage and event type: risk increased for AHIREM and AHITST, while it stayed the same or decreased for AHINREM and most central apneas.”
Introduction
•    Lines 73-81: The paragraph on AHI subtypes disrupts flow. Consider moving to Methods.

Response: We moved to methods as recommended.
2. Technical Corrections
Methods
•    Line 94: "8:00 pm" should be "20:00" for consistency.

Response: changed to 20:00
•    Lines 106-114: The definitions of comorbidities need standardization (ICD codes preferred).

Response: we added: “The predominant causes of death over the 15 years of follow-up were cardiovascular (284, 33.3%), cancer (226, 26.5%), pulmonary (102, 12%). Based on the International Classification of Diseases (ICD-10), the main causes included: chronic heart failure I.50 (97, 21.1%), myocardial infraction I.21 - I.25 (72, 15.6%), stroke, intracerebral hemorrhage I.60 – I.64 (62, 13.5%) and chronic obstructive pulmonary disease J.44 (52, 11.3%). The detailed ICD-10 diagnosis codes were not included in this study; however, they may be provided upon reasonable request to the corresponding author. “
•    Line 150: "attained" should be "defined as."

Response: changed to defined as
Tables and Figures
•    Table 1: "sie.35" appears to be a typo (should be ~8.35 based on context).

Response: changed to 8.35
•    Table 2: "OS_TIME_YEARS" header needs clarification.

Response: we performed several changes
•    Figure 1: The flowchart is referenced but appears to be missing or mislabeled.

Response: Figure 1. Provides a schematic overview of the study design, including the applied exclusion criteria and the classification of causes of death according to ICD-10 codes
3. Statistical Reporting
•    Line 165: Specify R package versions for reproducibility.

Response: The main R packages and versions were:

– survival 3.8.3

– finalfit 1.1.0

– ranger 0.17.0

– ggplot2 4.0.0

– dplyr 1.1.4

– ggfortify 0.4.19

– tidyverse 2.0.0

– openxlsx 4.2.8

– forcats 1.0.0

– survminer 0.5.0

– readxl 1.4.5

– dynamichazard 1.0.2

We have added the following fragment to the Methods section explaining used package versions :

“Analyses used the following R packages (versions): survival 3.8.3, finalfit 1.1.0, ranger 0.17.0, ggplot2 4.0.0, dplyr 1.1.4, ggfortify 0.4.19, tidyverse 2.0.0, openxlsx 4.2.8, forcats 1.0.0, survminer 0.5.0, readxl 1.4.5, and dynamichazard 1.0.2.”
•    Lines 170-171: The log-rank test is mentioned for survival curves but results aren't clearly reported.

Response: As a visualization, we presented KM curves for patient groups. Additionally, we included a forest plot. We stated in methodology section (line 191-192)
•    Confidence intervals should be consistently formatted (e.g., "95% CI: 1.18-1.62" throughout).

Response: We have corrected the markings in a uniform manner.
4. Discussion Points
Interpretation Issues
•    Lines 261-264: The depression finding (protective effect) needs more cautious interpretation. Could this reflect survivorship bias or treatment effects?

Response: added “Patients with depressive disorders may demonstrate heightened concern for their overall health and increased sensitivity to alarming symptoms. This phenomenon is not restricted to the domain of mental health but may also extend to somatic conditions. Such individuals are often more vigilant toward bodily sensations, which may lead to earlier recognition of symptoms and proactive health-seeking behaviors. Several mechanisms may underlie this tendency, including somatization, health-related anxiety, and heightened interoceptive awareness commonly observed in depression. While this vigilance may facilitate earlier detection and intervention in the course of various diseases, it may also contribute to increased healthcare utilization and a potential overestimation of health risks. Future studies should further investigate the extent to which depressive symptomatology influences health perception and the timing of medical intervention across different patient populations.”

Round 2

Reviewer 2 Report

Comments and Suggestions for Authors

Authors of this manuscript have adecuately addressed the main issues raised in the first review round. They have clarified the rolo of age as a potential confuonder, indicating that it woas significant in the univariable analysis but not in the multivariable model, which is methodologically acceptable. Nevertheless, it might be helpfuf if age were explicitly mentioned in the main table or the supplementary material to avoid future misunderstandings.

The concern regarding treatmente and ethical aspects has been properly resolved by including a satatement confirming that all patients received standard therapeutic recommendations after polysomnography, in line with current guidelines. This clarification removes the initial doubt about untreated OSA over a long follow-up period. 

The discussion of the "protective" effect of depression has also been improved an d is now more balanced, suggesting a possible behavioral or residual confounding explanation rather than a true protective phenomenon. 

Finally, the authors expanded the discussion with references to comparable long-term studies and emphasized the scarcity of literature using hazard ration models, which strengthengs the novelty of their approach. 

Overall, the revised version is substantially improved, clearer, and methodologically solid. The responses are polite and scientifically coherent. I have no further major concerns. 

Author Response

We appreciate the reviewer’s comment. We performed an analysis of the hazard ratio for age and added it to Table 2 and the Appendix. In addition, we included the following statement in the text: OSA severity (mild, moderate, severe) was modeled with no-OSA as the reference. Variables were grouped a priori into four sets: (1) age, (2) clinical signs/symptoms and anthropometrics (weight, height, BMI, blood pressure); (3) comorbidities and medications; and (4) polysomnographic indices (see Table 3).

Reviewer 3 Report

Comments and Suggestions for Authors

Overall it's ok now.

Author Response

Thank you for the positive reception of all our responses to the submitted comments. We have added one more sentence: Counts of participants by follow-up interval (based on OS_time_years) were: 0–5 years: 1041 (25.9%), >5–10 years: 1911 (47.5%), and >10–15 years: 1071 (26.6%), which the reviewer requested but we inadvertently missed in the previous version.
